# Bioinspired rotary flight of light-driven composite films

Dan Wang[1,2,7], Zhaomin Chen[1,7], Mingtong Li [ID][3,7], Zhen Hou[1], Changsong Zhan[1], Qijun Zheng[4], Dalei Wang[1], Xin Wang[1], Mengjiao Cheng [ID][2], Wenqi Hu [ID][3], Bin Dong [ID][1] ✉, Feng Shi [ID][2] ✉ & Metin Sitti [ID][3,5,6] ✉

Light-driven actuators have great potential in different types of applications. However, it is still challenging to apply them in flying devices owing to their slow response, small deflection and force output and low frequency response. Herein, inspired by the structure of vine maple seeds, we report a helicopter-like rotary flying photoactuator (in response to 0.6 W/cm² near-infrared (NIR) light) with ultrafast rotation (~7200 revolutions per minute) and rapid response (~650 ms). This photoactuator is operated based on a fundamentally different mechanism that depends on the synergistic interactions between the photothermal graphene and the hygroscopic agar/silk fibroin components, the subsequent aerodynamically favorable airscrew formation, the jet propulsion, and the aerodynamics-based flying. The soft helicopter-like photoactuator exhibits controlled flight and steering behaviors, making it promising for applications in soft robotics and other miniature devices.

Inspired by invertebrate animals and plants, significant efforts have been devoted in recent years to the development of soft actuators[1–4]. In contrast to the rigid counterparts consisting of rigid components, which are inspired by vertebrates or organisms with exoskeletons, soft actuators are not only easy to deform and physically adaptable to the changes in their environment but also safe to interact with humans. Their flexibility, adaptability and reconfigurability, together with their infinite passive degrees of freedom, make the soft actuators particularly promising for different real-world applications, including soft robotics[5–7], sensors[8], soft electronics[9], surgery[10], drug delivery[11], prosthesis[12] and artificial muscles[13,14]. Among different soft actuators, those driven by light have recently drawn more and more attentions. Light-driven actuators could be remotely and precisely actuated, which are facilely realized by applying light with different intensities and wavelengths[15,16]. This advantage thus makes them particularly

attractive for soft robot[17], gripper[18,19], and biomimetic device[20] applications.

In order to achieve robust actuation while keeping high maneuverability, a variety of advanced light-responsive materials, which range from polyelectrolyte hydrogels[21], carbon-based materials[22–25], crystals[26] to shape-memory polymers[27,28], liquid-crystalline polymers[29,30] and low phase temperature materials[31,32], have been developed. Under the influence of external light signals, these engineered materials can directly convert the light stimulation into their structure/macroscopic property changes based on the forces/torques generated via physical/chemical reactions or their combinations. Interestingly, these changes could result in various locomotion modes, which include walking[33,34], crawling[35–37], rolling[38,39], jumping[40,41] and swimming[31,42]. Although significant progress has been made to improve the response of photoactuators (Supplementary Table 1), it is

[1]Institute of Functional Nano & Soft Materials (FUNSOM), Jiangsu Key Laboratory for Carbon-Based Functional Materials & Devices, State and Local Joint Engineering Laboratory for Novel Functional Polymeric Materials & Joint International Research Laboratory of Carbon-Based Functional Materials and Devices, Soochow University, Suzhou, Jiangsu 215123, China. [2]State Key Laboratory of Chemical Resource Engineering, Beijing Laboratory of Biomedical Materials & Beijing Advanced Innovation Center for Soft Matter Science and Engineering, Beijing University of Chemical Technology, Beijing 100029, China. [3]Physical Intelligence Department, Max Planck Institute for Intelligent Systems, 70569 Stuttgart, Germany. [4]Department of Chemical Engineering, Monash University, Clayton, VIC 3800, Australia. [5]Institute for Biomedical Engineering, ETH Zürich, 8092 Zürich, Switzerland. [6]School of Medicine and College of Engineering, Koç University, 34450 Istanbul, Turkey. [7]These authors contributed equally: Dan Wang, Zhaomin Chen, Mingtong Li. ✉e-mail: bdong@suda.edu.cn; shi@mail.buct.edu.cn; sitti@is.mpg.de

still challenging to achieve the flying locomotion mode for the photoactuators because of their slow response speed, small actuation force output and low response frequency.

In this paper, we report a rotary flying photoactuator which is inspired by the vine maple seed. It features an ultrafast rotation with a speed as high as ~7200 revolutions per minute (rpm) and a fast response time as short as ~650 ms upon light stimulation. The superior flying performance is achieved by harnessing the synergistic interactions between the photothermal graphene and the hygroscopic agar/silk fibroin constituents and the following coordinated airscrew formation, jet propulsion and aerodynamics-based flight. The rotary flying helicopter-like photoactuator exhibits the well-controlled motion behavior and direction so that it could not only mimic the wind-dispersal behavior of the vine maple seed but also fly over a barrier or across a trench. The photoactuator reported in this study may be used in various applications. As a proof-of-the-concept example, we have demonstrated its potential use in collective environmental monitoring.

## Results

### Fabrication and characterization of the rotary flying photoactuator

The photoactuator consists of graphene nanoplatelet, agar and silk fibroin. Graphene is selected because of its light weight, sheet-like structure and excellent photothermal property, while agar and silk fibroin are chosen due to their favorable hygroscopic properties (agar and silk fibroin)[43,44] and adhesive nature (silk fibroin)[45], which could not only facilitate the light actuation process but also help to maintain the integrity of the photoactuator. The photoactuator has a typical size of 10 mm × 2 mm × 60 μm (length × width × thickness) with microchannels (with depth and width of 7.5 μm and 50 μm, respectively) on its surface (Fig. 1b and Supplementary Fig. 1), which is patterned by the template method (Fig. 1a). Note that the surface microchannels are crucial for the shape deformation of the photoactuator. The photoactuator has a layered cross-sectional structure, as can be seen from the scanning electron microscopy (SEM) images shown in Fig. 1c, d, which may be formed due to the π-π interactions[46] between the graphene nanosheets. Also, the photoactuator has a relatively hydrophilic surface (the water contact angle is 65.2°, as shown in Supplementary Fig. 2) with a root-mean-square roughness (Rq) (determined by atomic force microscopy (AFM), Fig. 1e and Supplementary Fig. 3) of ~163 nm. The Young's modulus (Supplementary Fig. 4) of the photoactuator is 7 MPa. The constituents inside the composite film are characterized by both the energy-dispersive X-ray (EDX) analysis and Fourier transform infrared spectroscopy (FTIR). Among others, the EDX analysis (Supplementary Fig. 5) shows that the sulfur elements (characteristic elements from the silk fibroin) distribute throughout the whole photoactuator, indicating the presence of silk fibroin. In addition, as indicated by the FTIR spectra (Supplementary Fig. 6), the characteristic peaks originating from graphene (blue curve, bands at 1400 cm$^{-1}$ and 1628 cm$^{-1}$ can be attributed to the bending vibration of C-OH bonds and the stretching vibration of C=O bonds, respectively), agar (red curve, the intense band at 1059 cm$^{-1}$ may be assigned to the coupling of the C-O/C-C stretching modes with the C-O-H bending modes of the polysaccharide), and silk fibroin (black curve, band at 1562 cm$^{-1}$ can be ascribed to the C=O stretching) can be observed in the graphene/agar/silk fibroin photoactuator film (green curve), thus confirming the presence of graphene, agar and silk fibroin components. Note that the shift of the band from 3403 cm$^{-1}$ in case of the graphene/agar composite (purple curve, which is the absorption band of hydroxyl) to 3386 cm$^{-1}$ in the case of the graphene/agar/silk fibroin composite (green curve) shows the interaction between different components through hydrogen bonding in the presence of silk fibroin, which lays the foundation for the film's ductility[47]. Moreover, graphene/agar/silk fibroin shows a wide UV-Vis-NIR absorption peak in the

whole spectrum region (250–1100 nm, Supplementary Fig. 7) which is likely due to the light absorbance of the graphene constituent. This indicates that the photoactuator film may be able to absorb the light energy and convert it into the thermal energy, which thus paves the way toward its photoactuation based on the photothermal effect.

### Rotary flight of the photoactuator

The photoactuator could be airborne in a way similar to a helicopter (Fig. 1f). When irradiating the surface of the photoactuator film, it rotates and flies, as shown in Fig. 1g–k and Supplementary Movie 1. The rotary flight of the helicopter-like photoactuator exhibits a fast response time and a take-off speed of ~650 ms and ~0.76 m/s (Fig. 1l), respectively (under 0.6 W/cm$^2$ NIR irradiation). There are two notable features for the current rotary flying photoactuator. One is the ultrafast rotational speed which can be visualized by the motion blur indicated by the circle shown in Fig. 1i and Supplementary Movie 1. Through motion analysis, the rotational motion is estimated to be as fast as ~7200 rpm (Fig. 1m). The ultrafast rotation in the current study thus lays the foundation for the flight behavior. It is in sharp contrast to the previously developed photoactuators which have relatively low rotational speeds (no more than 300 rpm), making them difficult to generate sufficient lift force for flying (Supplementary Table 2). The rotational motion in the current study represents the fastest rotation speed reported to date for light-driven rotary actuators (Supplementary Table 2). The other feature is the involvement of the aerodynamics. As can be seen from Fig. 1n, the trajectory of the helicopter-like photoactuator can be divided into three regions, climbing (with ascending vertical and horizontal speeds of 0.38 m/s and 0.69 m/s, respectively, blue curve), forward flight with negligible change in the flying height (with a forward speed of 0.57 m/s, black curve) and descent (with descending vertical and horizontal speeds of 0.28 m/s and 0.51 m/s, respectively, red curve), which is in contrast to that of the parabolic movement. Among others, forward flight (Fig. 1n, black curve) along the horizontal direction indicates the involvement of the aerodynamics. The helicopter-like photoactuator could achieve a flying height as high as 1.3 cm and a distance as far as 6.5 cm.

### Mechanism of the rotary flight of the photoactuator

The rotary flight of the photoactuator possibly relies on the synergistic interplay between the photothermal graphene and the hygroscopic agar/silk fibroin and the sequential gasification, airscrew formation, jet propulsion, and aerodynamics-based flight (Fig. 2a–f). We have studied the actuation mechanism by recording the actuation process using a high-speed camera, revealing the corresponding temperature variation by thermal imaging and computer simulation. Among others, as indicated in the high-speed camera images shown in Fig. 2g–i and the corresponding movie shown in Supplementary Movie 2, the photoactuator starts twisting within 40 ms which is guided by the microchannels (Fig. 1b) on its surface (i.e., twisting along the direction perpendicular to the microchannels). At ~645 ms, an elliptically-shaped protrusion, which is off the center of the film, is formed at the irradiated position (Fig. 2i). The protrusion is hollow, as evidenced by the empty interior of the sectioned protrusion (Supplementary Fig. 8). And the protrusion only expands to one edge of the graphene/agar/silk fibroin photoactuator film while the opposite edge remains intact (Supplementary Fig. 8). We have monitored the temperature variations during the shape deformation process. As shown in Fig. 2k–n, the temperature of the photoactuator film increases from 25.5 °C to 165.4 °C (beyond the boiling point of water (100 °C)) within 650 ms under light irradiation. This rapid rise in temperature is likely due to the photothermal effect of graphene, which could lead to the gasification of water inside the photoactuator film, resulting in the protrusion formation (Fig. 2c). Additionally, accompanied by the protrusion formation, the photoactuator film further twists (Fig. 2i and Supplementary Movie 2) due to the internal tension caused by the protrusion.

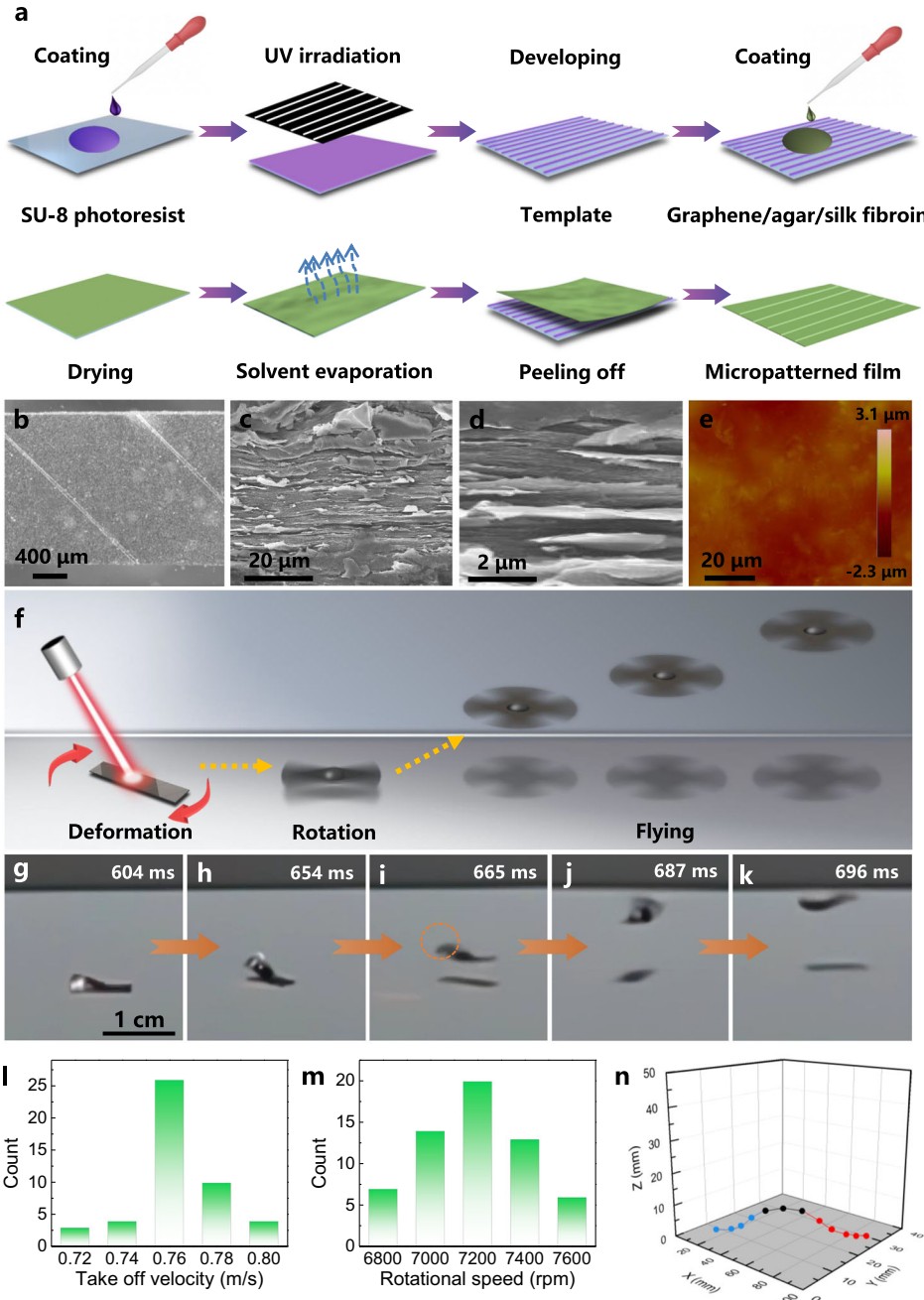

**Fig. 1 | The composite film-based photoactuator and its rotary flight capability.** **a** Fabrication process of the graphene/agar/silk fibroin photoactuator film with surface microchannels. **b** Optical microscope images showing the microchannels on the surface of the photoactuator. **c** SEM and (**d**) magnified SEM images showing the layered cross-sectional structure of the graphene/agar/silk fibroin photoactuator film. **e** AFM image showing the surface morphology of the graphene/agar/ silk fibroin photoactuator film. **f** Schematic showing the light-driven rotary flight behavior. **g**–**k** Time-lapse photos showing the take-off process of the photoactuator. **l** The take-off velocity of the photoactuator. **m** Rotational speed of the photoactuator during the take-off process. **n** Example 3D flight trajectory of the helicopter-like photoactuator.

The combined film twisting and protrusion formation can generate a unique airscrew-like structure similar to that of the vine maple seed (Fig. 2j). It is worth pointing out that the maximum strain of the composite film during the airscrew-like structure formation process is estimated to be around 16%, which is lower than the fracture strain of the photoactuator film (22.5%, Supplementary Fig. 4). As the protrusion continues to expand and reaches the film edge, the water vapor could escape from the edge, leading to the jet propulsion. When irradiating a fixed photoactuator film, the generated jet propulsion can be directly visualized by the blown white powders which are originally placed beside the film (Supplementary Fig. 9 and Movie 3). Additionally, we have examined the photoactuator film after the actuation process. As shown in Supplementary Fig. 10, the structural changes of the protrusion surface and the film edge where the vapor escapes are negligible. This is possibly because the water vapor escapes from between the lateral laminae (Fig. 2e) and the strong hydrogen-bonding-based (Supplementary Fig. 6) adhesion force could help to keep the photoactuator film intact. The position where the vapor escapes is further verified by the split formation in the case of the graphene/agar film (without the addition of silk fibroin) after the jet propulsion process (Supplementary Fig. 11). This control experiment also demonstrates that

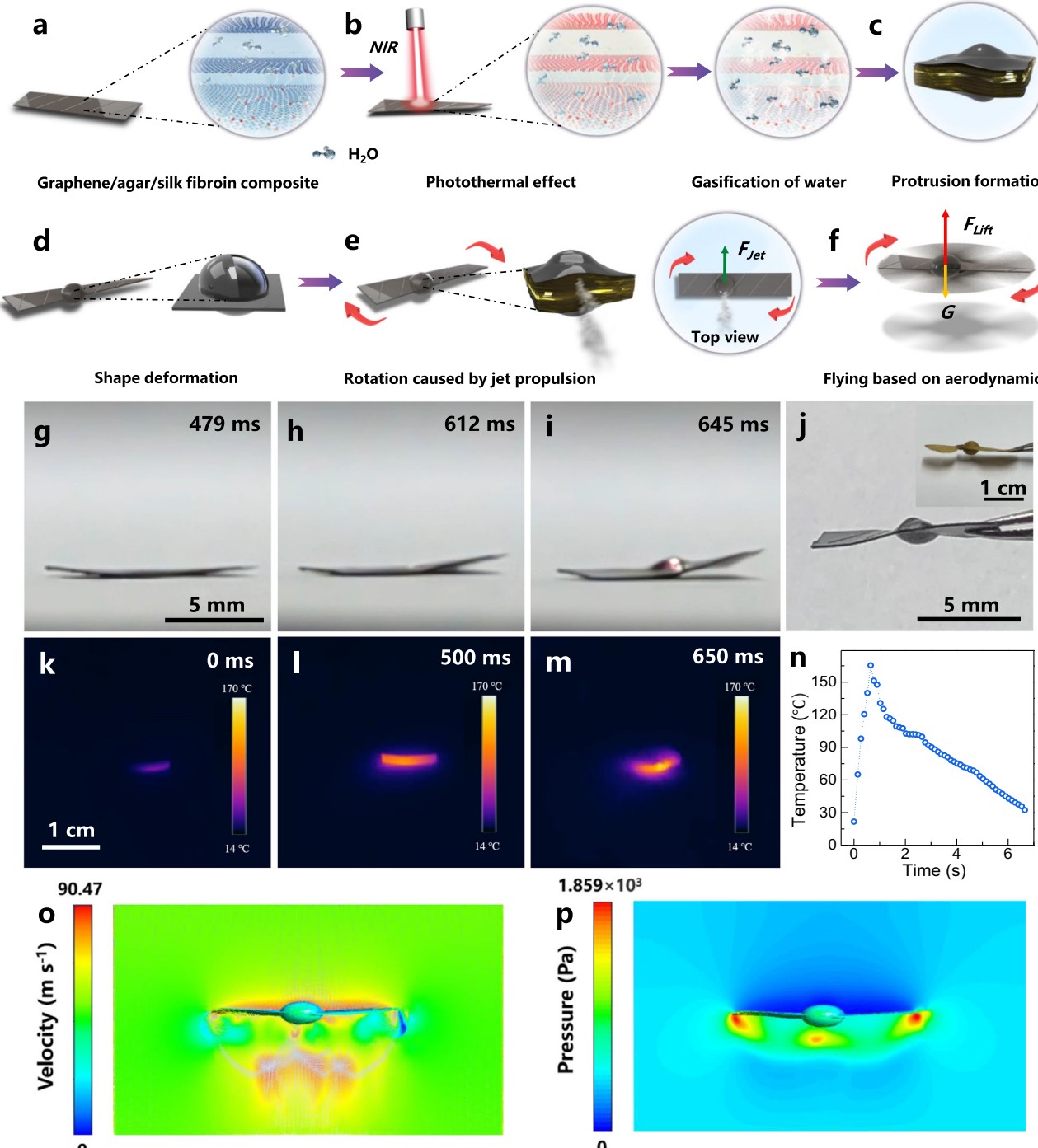

**Fig. 2 | Flight mechanism of the helicopter-like photoactuator. a–f** Schematic illustrating the flight mechanism of the helicopter-like photoactuator. **g–i** High-speed camera images showing the formation of the airscrew structure. **j** Photo of the resulting airscrew structure (inset: photo of a winged vine maple seed). **k–m** Infrared thermal images and (**n**) the corresponding temperature changes of the photoactuator. The CFD module in ANSYS Fluent simulation showing (**o**) the velocity and (**p**) pressure field distribution surrounding a high-speed rotating photoactuator with the airscrew structure.

the silk fibroin is essential to maintain the integrity of the photo-actuator film after actuation. Since the jet propulsion position is off the mass center of the photoactuator film (Fig. 2e), the thrust could generate a rotational force. The fast rotation of the twisted airscrew-like film in air generates the aerodynamic force and lifts the film into the air (Fig. 2f). The involvement of the aerodynamic lift is further confirmed by the control experiment (i.e. the vacuum experiment). As shown in Supplementary Fig. 12 and Movie 4, the airscrew-like photoactuator film is incapable of being airborne

under light irradiation when placed inside a vacuum environment (0.075 torr).

According to the above observations, we propose the following possible mechanism. Under the influence of the rapidly increased temperature due to the photothermal effect of graphene and the guidance of the surface microchannels, the photoactuator film undergoes the shape deformation, forming the airscrew-like structure with a protrusion at the irradiation spot. As the elliptically shaped protrusion progresses to the film edge, the vaporized water escapes

from between the lateral laminae, causing the jet propulsion and the following ultrafast rotation. This, when combined with the airscrew-like structure, results in the generation of the aerodynamic force and the following helicopter-like flight. This mechanism also implies that, in order to achieve the rotary flight of a photoactuator, it is desirable to satisfy the following features: 1) sufficient propulsion force output, 2) high frequency response, and 3) the aerodynamically favorable structure.

We have analyzed the lift force ($F_{lift}$) based on the Newtonian equation as:

$$F_{lift} = ma + mg \qquad (1)$$

where $m$ is the mass of the film (5.3 mg), $g$ is the acceleration of gravity (9.81 m/s$^2$), and $a$ is the vertical take-off acceleration of the film which is calculated to be 17.3 m/s$^2$ based on the following equation:

$$a = \frac{v - v_0}{t} \qquad (2)$$

where $v$, $v_O$ and $t$ are the vertical velocity of the photoactuator film (0.38 m/s), the initial velocity (0 m/s) and the time (approximately 22 ms to reach 0.38 m/s), respectively. The lift force ($F_{lift}$) can thus be calculated to be $1.44 \times 10^{-4}$ N based on Eq. 1.

Furthermore, we have developed a three-dimensional (3D) airscrew model and simulated the velocity and pressure distribution fields surrounding the fast-rotating airscrew based on the computational fluid dynamics (CFD) module in ANSYS Fluent. As can be seen from Fig. 2o, the air flow velocity increases above the high-speed rotating airscrew, which leads to the decrease of the local pressure (Fig. 2p). As a consequence, a pressure gradient is generated which provides the lift force for the rotational flying. The lift force ($F_{lift}$) can be estimated as:[48]

$$F_{lift} = \frac{1}{2} C_L \rho (\Omega R)^2 S \qquad (3)$$

where $\rho$ is the air density (1.2 kg/m$^3$ at 25 °C), $\Omega$ is the rotational velocity of the photoactuator film (754 rad/s), $R$ is the radius of rotation of the film (5 mm), and $S$ is the rotation area ($S = \pi R^2 = 7.85 \times 10^{-5}$ m$^2$), $C_L = 0.61$ is the lift coefficient which is calculated from the CFD module (the detailed CFD calculations are shown in Supplementary Figs. 13, 14 and the descriptions therein). The lift force can thus be calculated to be $4.12 \times 10^{-4}$ N, which is in the same order of magnitude as the lift force calculated based on Eq. 1 ($1.44 \times 10^{-4}$ N), thus demonstrating that the rotation-induced lift force is likely to be the driving force for the flight.

Moreover, we have monitored the mass changes of the photo-actuator film before and after light-driven flight. As shown in Supplementary Fig. 15, the weight of the photoactuator film after actuation decreases by approximately 9.5 wt%, indicating that some of the water inside the photoactuator film has been released. Note that the water content inside the photoactuator film is around 12 wt%, as determined by thermogravimetric analysis (TGA) shown in Supplementary Fig. 16. In addition, the degradation temperature of the photoactuator film is 230 °C (Supplementary Fig. 16), indicating that it is thermally stable during the actuation process (165.4 °C, Fig. 2n). Due to the hygroscopic nature of the agar/silk fibroin components, the lost water could be replenished[43,44]. As can be seen from Supplementary Fig. 15, the weight of the photoactuator film could be recovered to the original value when exposing the film to the humidity environment (90% relative humidity). This, together with the negligible changes in the morphology and structure of the actuator before and after the light-driven actuation, allows the helicopter-like photoactuator to be reused for at least 7 times (Supplementary Fig. 17). We have also examined the water

re-absorption time after light actuation for 7 cycles. As shown in Supplementary Figs. 18, 19, it takes approximately 5.5 min to re-absorb water after repetitive light actuation and the local heating does not affect the capability of the film to re-absorb water. In addition, it is worth pointing out that the local heating could generate a preferred spot at which the jet propulsion occurs upon repetitive light actuation (Supplementary Figs. 20–22).

## Motion control of the rotary flying helicopter-like photoactuator

In this section, we have first studied the influence of the material composition, size and thickness of the photoactuator film and irradiation light intensity on the rotational speed, the flying height, and the landing distance of the helicopter-like photoactuator, respectively. Then, we realize the control over the rotational direction, the flying height, and the flying direction of the rotary flight of the photoactuator by adjusting the jet propulsion position, angle of attack and elevation angle, respectively.

The rotary flight of the helicopter-like photoactuator relies on the fast rotation of the airscrew-like structure. The critical components inside the photoactuator film that determine the rotational speed are graphene and water. Among others, the graphene content has to be high enough (>1 wt%) to ensure the sufficient photothermal conversion to activate the flying motion. But when the graphene content is too high, the water absorbing capability is reduced due to the increased hydrophobicity of the photoactuator film, which leads to the decrease of the rotational speed, the flight height, and the landing distance (Supplementary Fig. 23). The optimal graphene content is around 4.6 wt%. In addition, the helicopter-like photoactuator containing less $H_2O$ could only form a small protrusion due to the limited amount of water vapor generation upon irradiation. If the $H_2O$ content is too high, the excess water decreases the heating efficiency caused by the photothermal graphene. Therefore, either less or more $H_2O$ content results in the decrease of the rotation speed, flight height and landing distance (Supplementary Fig. 24). An intermediate $H_2O$ content of 12 wt% is thus preferable. Furthermore, we have evaluated the effect of thickness and size of the photoactuator film. The thicker or larger film will increase the weight, resulting in the decreased rotational speed, flight height and landing distances (Supplementary Figs. 25, 26). While for the thinner photoactuator film (thickness less than 50 μm), no protrusion formation could be observed and the film bends instead under light irradiation. And for the smaller-sized photoactuator film, the protrusion is also smaller, which could not provide sufficient propulsion for the fast rotation. The optimum graphene content, $H_2O$ content, thickness and size are thus 4.6 wt%, 12 wt%, 60 μm and 10 mm × 2 mm (length × width), respectively. Additionally, the flying performance of the helicopter-like photoactuator also highly depends on the light intensity. As can be seen from Supplementary Fig. 27, the rotational speed, flying height and landing distance gradually increase with the increasing light intensity. This is because the higher light intensity could generate more heat through photothermal conversion, leading to stronger jet generation. Note, when the light intensity is higher than 0.7 W/cm$^2$, the photoactuator burns.

For aerodynamics, the angle of attack is known to have a great influence on the flight behavior[49]. In the current study, the flying height could be controlled by adjusting the angle of attack of the helicopter-like photoactuator. Note that different angle of attack (i.e., the angle between the direction of the relative wind and the chord of an airfoil ($\beta$), as shown in Fig. 3a) of the helicopter-like photoactuator could be realized by controlling the alignment angle of the microchannels ($\alpha$, with respect to the short axis of the photoactuator film, Supplementary Fig. 28). As shown in Fig. 3b, the flight height is the highest when the angle of attack is approximately 15°. The lift force increases with the increasing angle of attack when the angle is lower than 15° due to the increased pressure difference between the lower and upper surface of

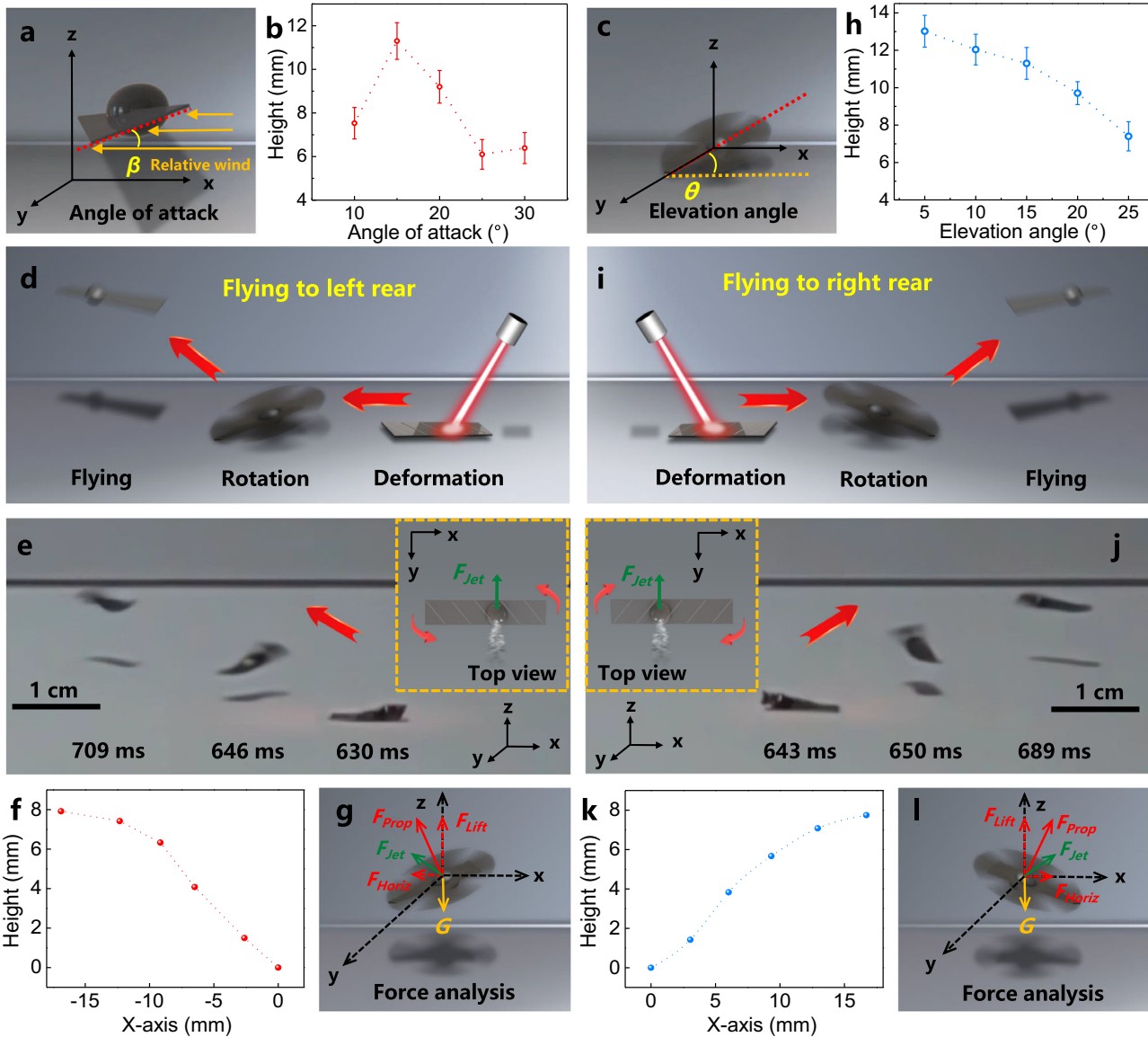

**Fig. 3 | The flight motion control of the helicopter-like photoactuator.**
**a** Schematic illustration showing the angle of attack (β) of the airscrew structure.
**b** The effect of the angle of attack on the flight height. **c** Schematic showing the
elevation angle (θ) of the helicopter-like photoactuator. **d** Schematic illustration,
(**e**) overlaid CCD images, (**f**) the motion trajectory and (**g**) the corresponding force
analysis showing the helicopter-like photoactuator flying to the left rear, which is
realized by controlling the elevation angle. The images are captured from Sup-
plementary Movie 6. **h** The influence of elevation angle on the flight height.
**i** Schematic illustration, (**j**) overlaid CCD images, (**k**) the motion trajectory and (**l**)
the corresponding force analysis showing the helicopter-like photoactuator flying
to the right rear, which is realized by controlling the elevation angle. The images are
captured from Supplementary Movie 7. Note that the distance to the left is defined
as negative. The top view schemes indicating the driving force and rotational
direction are shown in the insets in (**e**) and (**j**), respectively. $F_{prop}$, solid red arrow, is
the propulsion force generated as a result of the fast rotation. $F_{horiz}$, dashed red
arrow, represents the horizontal component force. $F_{lift}$, dashed red arrow, is the
vertical component force (lift force). G, solid yellow arrow, is the gravitational force
of the photoactuator. Error bars denote the standard deviation.

the airscrew, which is caused by the increased rate of air flow on the
upper surface of the film. However, the air flow separation may occur
on the upper surface of the film after a certain angle of attack (>15°),
resulting in the increased pressure on the upper surface and the
decreased lift force due to the decreased pressure gradient. This result
is consistent with the simulations (Supplementary Figs. 29, 30).

Furthermore, the clockwise and counterclockwise rotational
direction could be controlled by the position where the jet propulsion
occurs. Among others, when the jet propulsion occurs at the lower left
part of the photoactuator film (Supplementary Fig. 31a–d), the film
rotates clockwise. The counterclockwise rotation could be realized
when the jet propulsion occurs at the lower right part of the photo-
actuator film (Supplementary Fig. 31e–h). Note that, for the helicopter-
like photoactuator shown in Supplementary Fig. 31e–h, when the

rotational direction is reversed (from clockwise rotation to the coun-
terclockwise one by controlling the location of the jet propulsion), the
angle of attack will become negative so that the photoactuator could
only exhibit the rotational behavior instead of the rotary flying motion
because of the higher pressure on the upper surface of the airscrew
with respect to the lower surface (Supplementary Fig. 32 and Supple-
mentary Movie 5). In addition, we find out that, by changing the
alignment angle (α) to −45°, the counterclockwise rotation could result
in flying motion because of the positive angle of attack in this case
(Supplementary Fig. 33).

The elevation angle is critical to determine the flying performance
of a helicopter. In the current study, the flight direction, flying height
and landing distance could all be controlled by the elevation angle (the
angle between the plane of rotation and the horizontal plane (θ),

Fig. 3c). The adjustment of the elevation angle could be realized by adjusting the position of the protrusion caused by the irradiation spot. For example, when irradiating the right side of the photoactuator film which is placed along the x-axis (Fig. 3d and Supplementary Fig. 34), the formed protrusion divides the film into two parts, i.e., the short airscrew part and the long airscrew part. The long airscrew which is heavier touches the substrate and the whole airscrew structure thus faces the left. As a consequence, the jet propulsion leads to the flight toward the left rear direction (Fig. 3e, f and Supplementary Movie 6) because of the horizontal component force ($F_{horiz}$, which is toward the left) of the propulsion force ($F_{prop}$, Fig. 3g solid red arrow, which is generated as a result of the fast rotation). Note that the motion direction deviates toward the rear which is likely because the jet propulsion ($F_{jet}$, Fig. 3g solid green arrow) which is in the xy-plane as evidenced by the rotation only motion under vacuum (Supplementary Fig. 12) also pushes the airscrew-like structure backward. The elevation angle could be well controlled by controlling the position of the protrusion (Supplementary Fig. 34). As a result, the flying height and distance could also be regulated, i.e., the flight height decreases (Fig. 3h) and the landing distance increases (Supplementary Fig. 35a) with the increasing elevation angle. This is likely due to the reduced vertical lift component force ($F_{lift}$) and increased horizontal component force ($F_{horiz}$) with the increasing elevation angle (Supplementary Fig. 35b, c). On the other hand, when irradiating the left half of the photoactuator film, the airscrew-like structure faces the right so that it could fly toward the right rear direction upon light actuation (Fig. 3i–l and Supplementary Movie 7).

## Potential application demonstration of the rotary flight of the helicopter-like photoactuator

In this section, we have first studied the wind-dispersal behavior of the photoactuator and compared it to that of the plant counterpart. Second, we have examined the parameters that may influence the wind-dispersal behavior of the photoactuator. Third, as a proof-of-the-concept example, we have explored the potential of the photoactuators as wind-dispersal colorimetric sensors for collective environmental information collection. Fourth, we have also explored the potential of the photoactuator in obstacle crossing.

In nature, many plants use wind to spread their seeds. Due to the unique 3D shape, seeds can be wind-dispersed to a remarkable distance. As the shape of our photoactuator is similar to that of the vine maple seed (Fig. 4a–c), we have first fabricated a photoactuator with a size similar to that of the vine maple seed (80 mm²). Depending on the varieties of vine maple trees, their heights range from less than 1 m to over 4 m. The seeds of the vine maple tree in our campus distribute randomly on the tree with heights ranging from -0.5 m to -2.5 m (averaging 1.5 m). We thus choose to release the photoactuator at a similar height, i.e., 1.5 m (Supplementary Fig. 36). We have compared the rotary-falling process of the photoactuator with that of the winged vine maple seed. As can be seen from Fig. 4d–g and Supplementary Movie 8, the photoactuator exhibits the falling behavior with rotations (Fig. 4f, g), which is similar to that of the winged vine maple seed (Fig. 4d, e). The synchronous rotary-falling compared to that of the winged vine maple seed under similar circumstances is also the indication of the involvement of the aerodynamics. Interestingly, as illustrated in Supplementary Fig. 37 and Supplementary Movie 9, in the presence of crosswind (the wind speed is 3 m/s), the photoactuator shows apparent horizontal displacement which slows down the descent. We have evaluated the wind-dispersal distance of the photoactuator under different wind speeds. The photoactuator, which is light-actuated at a height of 1.5 m and subject to 3 m/s wind, could be wind-dispersed to a distance as far as 1.1 m (Supplementary Fig. 38 red column). This flying behavior is similar to that of the winged vine maple seed (the wind-dispersal distance of the vine maple seed is 1.05 m under the same circumstance). We have further compared the

wind-dispersal distance of the helicopter-like photoactuator with and without the airscrew-like structure. As shown in Supplementary Fig. 38, the wind-dispersal distance of the film with the airscrew-like structure (red column) is significantly farther than that without the airscrew-like structure (orange column), revealing the importance of the airscrew-like structure. This is because the film without the airscrew-like structure tumbles during the falling process (Supplementary Movie 10), while the composite film with the airscrew-like structure could maintain the stability during the flying motion. Furthermore, the release height and size of the photoactuator could also influence the wind-dispersal distance. As shown in Supplementary Fig. 39, the photoactuator can be wind-dispersed to a farther distance when the release height is increased, and the wind-dispersal distance can reach approximately 10 m when the release height is 5 m (the wind speed is 3 m/s). In addition, as can be seen from Supplementary Fig. 40, there is a maximum wind-dispersal distance (2.2 m) when the size of the photoactuator is about 20 mm² (1.5 m release height and 3 m/s wind speed). This is possibly because the photoactuator size should be large enough to generate the lift force. However, as the size increases, the weight of the photoactuator also increases, which may result in the decrease of the flying ability.

Inspired by nature, researchers have developed wind-dispersal microfliers with on-board sensors (colorimetric sensors (read out based on color analysis of the digital images) or electronic ones (read out through wireless links)), which could distribute themselves over a large area to collect collective information for environmental monitoring or sensing purposes[50,51]. However, the deployment of these microfliers normally relies on the manual operation. Due to the untethered feature of the light actuation method, the photoactuators may be activated from a remote distance, thus facilitating their controlled and on-demand release from a high altitude for wind-dispersal over a large area. As a proof-of-the-concept example, we have explored the potential of the photoactuators in collective colorimetric sensing for environmental monitoring. As shown in Supplementary Fig. 41a, the photoactuator is loaded with four thermosensitive gels and four humidity sensitive gels, which could be utilized to collect the temperature and humidity information, respectively. A transparent polydimethylsiloxane (PDMS) substrate is also attached onto the photoactuator in order to collect the particle matter for pollutant measurement. After light actuation and wind-dispersal, the information about temperature and relative humidity could be visually read out through the color changes of the thermosensitive and humidity sensitive gels (Supplementary Fig. 41b–e). The size of the particulate pollutant could also be obtained by analyzing the particle matter on the PDMS substrate (Supplementary Fig. 41f–h) based on the dynamic light scattering (DLS) method. Therefore, by combining the flying motion and the wind-dispersal capability of a number of photoactuators, it is possible to achieve the collective sensing in the complex environment.

To this end, we have set up four areas with different environmental characteristics, representing hot, cold, humid and dusty environments (Supplementary Fig. 42). To monitor the environmental characteristics of different areas, we have controlled the flight of eight photoactuators (Number 1 to 8) to fly toward different regions (the release height and the wind speed are 1.5 m and 3 m/s, respectively). By directly visualizing the attached colorimetric sensors on the photoactuators based on the captured digital image, we could simultaneously collect the environmental information from different regions. Additionally, it is also possible to evaluate the particulate pollutant level based on the DLS measurement. Based on this method, the distribution of temperature, humidity and size of the particulate pollutant (Supplementary Fig. 43) over the whole area could be obtained. Compared to the traditional sensors for environmental monitoring, the photoactuator based ones may offer several advantages: 1) The distributed collections of multiple sensors allow for simultaneous and

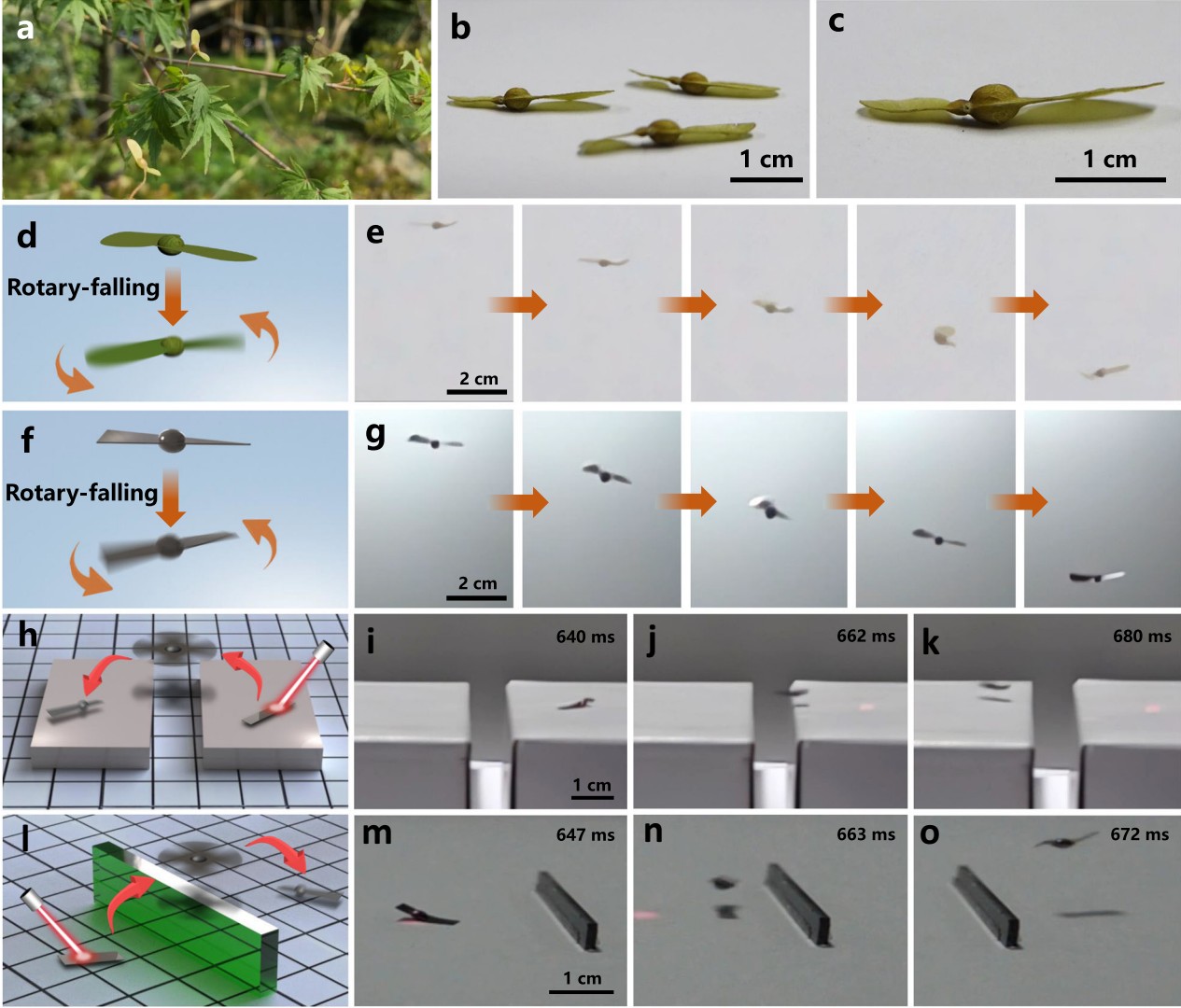

**Fig. 4 | Potential application demonstration of the helicopter-like photo-actuator. a–c** Photos of the vine maple seeds. Schematic illustration and the corresponding time lapse images within 66 ms captured from Supplementary Movie 8 showing the rotary-falling of (**d**, **e**) winged vine maple seed and (**f**, **g**) the helicopter-like photoactuator. Schematic illustration and the corresponding time lapse images captured from Supplementary Movies 11, 12 indicating the helicopter-like photoactuator (**h–k**) flying across a trench or (**l–o**) flying over a barrier.

collective sensing to obtain information from different areas; 2) The active light actuation enables the flying locomotion with the guided direction and the passive wind-dispersal could increase the flying distance, both of which make the photoactuators suitable for collective sensing; 3) The airscrew-like structure makes the flying stable without tumbling, which could also facilitate the distributed sensing.

In recent years, significant progress has been made in the field of small-scale (several hundred microns to several millimeters) robots[52], which hold great promises in diverse applications ranging from targeted cargo delivery[53], diagnosis to sensing[54] and drug evaluation[55]. In order to perform various tasks, such as explorations and inspections, it is required that the small-scale robot has the capability to overcome different obstacles. The fast response and the flight capability, together with the excellent controllability, render the current photoactuator applicability under different circumstances. As a proof-of-the-concept example, we have evaluated its potential in obstacle crossing. As shown in Fig. 4h–k and Supplementary Movie 11, the photoactuator can take off from the right platform, fly across a 11 mm wide trench, and then land on the left platform. The widest trench the current photoactuator can fly across is approximately 65 mm. In addition, the photoactuator could fly over a barrier under light actuation. As can be

seen from Fig. 4l–o and Supplementary Movie 12, the photoactuator can rapidly fly from one side of the barrier which is 6.3 mm in height to the other side. The maximum height of the barrier the photoactuator can overfly is around 11.3 mm, which is slightly lower than the highest flight height (13 mm). The fast passing and the obstacle-crossing ability make the current photoactuator adaptive to unstructured environments and attractive for various applications.

## Discussion

A bioinspired rotary flying helicopter-like photoactuator has been achieved with an ultrafast rotational speed of ~7200 rpm and a fast response time of ~650 ms. The actuation relies on the synergetic interactions between the photothermal graphene and the hygroscopic agar/silk fibroin components and the sequential aerodynamically favorable airscrew formation in conjunction with the jet propulsion. The rotational speed, flying direction, flight height and landing distance of the helicopter-like photoactuator could be well controlled by adjusting the irradiation intensity and the irradiation position. The unique motion behavior allows the photoactuator to function as a mimic of the wind-dispersal vine maple seed or fly over a barrier/across a trench. The proposed photoactuator may not only stimulate more

advanced design principles for the realization of the flying motion but also pave the way toward its application in the fields of soft robotics and other miniature devices. Specifically, it is anticipated that the photoactuator reported in the current study could be utilized for the distributed environmental sensing and information collection over a large area based on either high resolution aerial digital imaging or wireless links (through the integration of wireless modules).

## Methods

### Materials

Agar and N,N'-dimethylformamide were purchased from Acros Organics. Graphene nanoplatelets (5 μm in size, 6–8 nm thick) were purchased from Strem Chemicals, Inc. Silk fibroin ($M_w$ = 6–10 k) was purchased from Aladdin Co., Ltd. SU-8 2007 photoresist was obtained from Microchem, Newton, MA, USA. Silicon wafer was obtained from Topvendor Technology Co., Ltd. PDMS was purchased from Nanjing MKNANO Tech. Co., Ltd. The thermosensitive gel consisting of the leuco dye-developer (dodecyl gallate, etc.)-solvent (octadecanol, etc.) system embedded in the epoxy polymer matrix was obtained from Shenzhen MiJiang Tech. Co., Ltd. The humidity sensitive gel containing the silica gel and the color-changing indicator (i.e. cobalt chloride) was purchased from Dongguan Zhongqi Printing Equipment Co., Ltd.

### Preparation of the micropatterned silicon wafer template

The micropatterned silicon template was fabricated by the photo-lithographic method. Silicon wafer was first immersed in the piranha solution (the volume ratio of $H_2O_2$ to $H_2SO_4$ solution was 3:7) for 3 h, and then washed by deionized water and dried by nitrogen gas. SU8 photoresist was then spin-coated onto the pretreated silicon wafer (500 rpm for 10 s followed by 2500 rpm for 30 s). The photoresist coated silicon wafer was heated at 95 °C for 3 min. A patterned photomask was placed on top of it. After UV (14.5 mW/cm$^2$) irradiation for 10 s, the silicon wafer was heated at 95 °C for 3 min and then immersed in SU-8 developer for 3 min to dissolve the non-irradiated area. Finally, the micropatterned silicon template was washed by isopropanol, dried by nitrogen gas, and heated at 150 °C for 30 min.

### Preparation of the graphene/agar/silk fibroin photo-actuator film

In a typical experiment, 1 g of agar was dissolved in 10 ml of N,N'-dimethylformamide and stirred for 2 h at 110 °C. After the whole solution was cooled down to 40 °C, 60 mg of graphene nanoplatelets and 100 mg of silk fibroin were added to the mixture and vigorously agitated for 4 h. 2 mL solution containing graphene/agar/silk fibroin was spin-coated on the patterned silicon template at 1500 rpm for 40 s. After drying, the graphene/agar/silk fibroin film was peeled off from the template, cut into strips with different sizes and kept in an environment with 90% relative humidity.

### Characterizations

The SEM experiment was carried out on a Carl Zeiss Supra 55 scanning electron microscope with an EDX analysis attachment. Before SEM observation, the sample was sputtered with 8 nm thick gold. AFM measurements were carried out on a Bruker Dimension Scanning Probe Microscope. The microscopic images of the film were obtained by a Nikon Eclipse 80i microscope. CCD images and Movies were captured by the SONY DSC-RX10III digital camera. Tensile stress-strain curve was obtained on the INSTRON 5982 universal testing machine. The trajectory and velocity of the motion were obtained by analyzing the captured movies using the PhysVis software. The movement of the graphene/agar/silk fibroin photoactuator film was actuated by an 808 nm laser (Hi-Tech Optoelectronics Company) which was placed 6 cm away from the film. Unless otherwise mentioned, the graphene, agar, silk fibroin and $H_2O$ contents inside the photoactuator film were 4.6 wt%, 75.8 wt%, 7.6 wt% and 12 wt%, respectively. The size and

thickness of the film were 20 mm$^2$ (10 mm in width and 2 mm in length) and 60 μm, respectively. The ratio between length and width of the film was kept at 5:1 to ensure the airscrew structure formation. The irradiation intensity, the angle of attack and the elevation angle were 0.6 W/cm$^2$, -15° and -15°, respectively. TGA of the graphene/agar/silk fibroin composite film was carried out on a Mettler Toledo thermo-gravimetric analyzer. FTIR measurement was carried out on a Hyperion spectrophotometer (Bruker). The infrared thermal images were obtained by utilizing the FLIR-A300 camera (FLIR Systems Inc.). UV-Vis-NIR spectrum was obtained on a PerkinElmer LAMBDA 750 UV/Vis/NIR spectrophotometer. The contact angle was measured by utilizing a Dataphysics OCA 20 contact angle system. The size distribution of particulate pollution was measured by Malvern Zetasizer Nano ZS90. The aerodynamic simulation was conducted based on the ANSYS Fluent software.

## Data availability

The data that support the plots within this paper are available in the Source data file. Additional data available from authors upon request. Source data are provided with this paper.

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

## Acknowledgements

This work was supported by National Key Research and Development Program of China (Grant No. 2018YFE0306105) and National Natural Science Foundation of China (Grant No. 22173068). This work is supported by Suzhou Key Laboratory of Functional Nano & Soft Materials, Collaborative Innovation Center of Suzhou Nano Science & Technology, the 111 Project, Joint International Research Laboratory of Carbon-Based Functional Materials and Devices. It was also supported by the Priority Academic Program Development of Jiangsu Higher Education Institutions (PAPD), the Fund for Excellent Creative Research Teams of Jiangsu Higher Education Institutions and Suzhou Key Laboratory of Surface and Interface Intelligent Matter (SZS2022011). Also, it was supported by the National Science Foundation for Distinguished Young Scholars (51925301) (F.S.), Max Planck Society (M.S., W.H.), and the Fundamental Research Funds for the Central Universities (QNTD20; XK1902) (F.S.), Wanren Plan (wrjh201903) (F.S.), and Open Project of State Key Laboratory (sklssm2023) (F.S.).

## Author contributions

M.S., F.S., B.D., and Dan.W. conceived the presented idea and designed the research, and Dan.W., Z.C., M.L., Dalei.W., Z.H., and C.Z. performed all the experiments. Z.C. and Z.Q. carried out the CFD module in ANSYS Fluent simulation. Dan.W., X.W., W.H., and M.C. analyzed the data. M.S., F.S., B.D., Dan.W., M.L., and Z.C. wrote the paper. All authors discussed the results and commented the manuscript.

## Competing interests

The authors declare no competing interests.
