## [Peer Review File · Nature Communications]

Bioinspired rotary flight of light-driven composite filmsREVIEWER COMMENTS

Reviewer #1 (Remarks to the Author):

The actuators have potentials in the field of soft robotics and sensing applications. In order to actuate the actuators, external stimuli such as light, could be used. The photoactuator and its design could be mainly classified into two categories, photothermal actuators and the liquid crystal polymer based ones. These photoactuators could already realize different motion behaviors, such as walking, jumping, etc. which are basically based on the shape deformation of the actuator materials. The mechanism reported in this manuscript is a new process which generates a new motion (e.g. flying motion). The manuscript could be published if the authors address the following concerns:

1. The authors proposed that the jet comes out from the side of the film. Will the film slide instead of flying? What prevents the film from sliding?
2. Is there any reason for the material selection or the composite recipe in the manuscript? The authors use the graphene, agar, silk fibroin as the photothermal material, hygroscopic material, and adhesive material. Is there any specific reason for such choice?
3. Is the method proposed by the authors a generally applicable way to realize the flying motion? If so, what is the designing principle for the flyable photoactuator, the authors should provide more discussion.
4. The authors proposed that the previously developed photoactuator could not achieve the flying motion, but what is the reason/mechanism lies behind it? The authors should elaborate the reason/mechanism.
5. The author should give an explanations and descriptions in details about what other researchers have done in order to improve the response/time?
6. According to the mechanisms shown in the manuscript, the lift force should be a little different in the case of Fig. S30a and b. But the forces look likes the same, the authors should give a more reasonable scheme; And scale bars should be provided in Fig. S9c-j.
7. In the conclusion part, the authors should provide some future perspective based on the works they have done in the manuscript.

Reviewer #2 (Remarks to the Author):

In this manuscript, Wang et al report on the bio-inspired rotary flight of composite films, driven by a photoactuator mechanism. The composite consists of a hygroscopic agar/silk fibroin film that embeds graphene. Upon exposure to near-infrared light, water in the film evaporates, leaves the film from the side and thereby generates a jet propulsion that drives the spinning of the film (up to 7200 rpm). Due to the aerodynamic shape of the film, this results in a “helicopter-like” lift. By changing the angle at which the NIR light hits the film, the flight direction of the film can be directed. For a film with mm dimensions, the authors report heights and horizontal displacements in the centimetre-regime.

The work is original, the paper is well-written and the suggested mechanism for the flight behaviour is well supported by experiments. However, the following points need to be addressed prior to publication in Nature Comm:

Clarification of the mechanism: from the schemes in the main text, the driving force for the flight is not very clear. Including a scheme like Figure S26a in the main text would help to explain the left- vs righthanded rotation. Also, the role of the microchannels in the twisting of the sheet is not clarified in the schemes.

Reusing the films for multiple flights: The authors mention that the films can fly multiple times. However, the underlying mechanism relies on the evaporation of water upon heating of the film. Hence, the spinning stops when all water is evaporated – which is within a couple of seconds, implying that the films can only fly short distances. How long does it take for a film to re-adsorb water, such that it will fly again upon new exposure to NIR?

Also, the authors mention that they inspected the film after the actuation process. The morphologies of the protrusion surface and the film are shown to be unaffected. However, does the local heating of the film affect the capability of the film to re-adsorb water? Could it be that the local heating generates a “preferred spot” to create the jet propulsion opening when the film is re-exposed to NIR? This might affect the capabilities to steer the film during subsequent flights.

Minor point: The paper mentions nanocomposite elements, but is it really nanocomposite? The films appear to me as if it consists of multi-layered graphene embedded in the composite, without control at the nano-level (i.e. single sheet graphene).

Reviewer #3 (Remarks to the Author):

I generally liked the idea of the paper and found the experimental realization very funny, I assume the authors had a very enjoyable project.

The science seems solid and well-done.

The paper itself, however, is not really at a level required for Nature Communication. While parts of the manuscript are well-written, in correct English and clearly structured, others are not. The beginning of the introduction repeats phrased and wordings, without paying intention to grammar, the use of articles... It is clear that different people (with different linguistic skill levels) wrote the article.

In this form, it should not be published without a thorough language editing.

Also, I find the application part absolutely not convincing, why is it impressive that from 1.5m starting height in presence of wind the helicopters are distributed about 1.1 m?

We have carefully considered the reviewers' comments and revised the manuscript accordingly, detailed as follows:

Reviewer 1

Comments: The actuators have potentials in the field of soft robotics and sensing applications. In order to actuate the actuators, external stimuli such as light, could be used. The photoactuator and its design could be mainly classified into two categories, photothermal actuators and the liquid crystal polymer based ones. These photoactuators could already realize different motion behaviors, such as walking, jumping, etc. which are basically based on the shape deformation of the actuator materials. The mechanism reported in this manuscript is a new process which generates a new motion (e.g. flying motion). The manuscript could be published if the authors address the following concerns:

Answer: We are grateful for the comments and constructive suggestions from Reviewer 1. We have carefully addressed all the issues and made proper revisions to the manuscript. Modifications are highlighted with blue font.

Q1. The authors proposed that the jet comes out from the side of the film. Will the film slide instead of flying? What prevents the film from sliding?

Answer: We appreciate this thoughtful comment. The film will not slide instead of flying. Due to the preformed 'airscrew'-like structure, the film exhibits fast rotation when propelled by the jet. Although there is a component force along the horizontal direction (F_{horiz} , Fig. 3g red dashed arrow), the rapid rotation provides sufficient lift force for flying in the vertical direction

in air. Therefore, the rotation-induced lift force prevents the film from sliding (but F_{horiz} could
control the flying direction (Fig. 3g and 3l)). However, owing to the disappearance of the lift
force in the vacuum environment, the film slides instead of flying under this condition
(Supplementary Video 4).

**Q2.** Is there any reason for the material selection or the composite recipe in the manuscript?
The authors use the graphene, agar, silk fibroin as the photothermal material, hygroscopic
material, and adhesive material. Is there any specific reason for such choice?

**Answer:** Thank you for your comments. To ensure the flight behavior of the ‘helicopter’-like
photoactuator, we considered the following factors when selecting the materials: 1). The
material should be light enough which could reduce the weight of the actuator; 2). The material
should have good water absorbing ability so that absorbed water could be vaporized that may
be utilized to generate sufficient propulsion force; 3). To facilitate the jet generation while
keeping the reusability, it is desirable to introduce weak interactions between the components
inside the film, such as hydrogen bonding; 4). The protrusion formation and the jet propulsion
at the edge of the actuator film are crucial, and it is necessary to obtain a layered structure
inside the film.

Taking these factors into account, we choose graphene as the photothermal material
because of its light weight, sheet-like structure and excellent photothermal property. We choose
agar and silk fibroin based on their favorable water absorbing capability. In addition, the
introduction of silk fibroin could also facilitate the hydrogen bond formation between different
components, thus helping to keep the integrity of the film during the light actuation.

**Revision in the manuscript:** Page 4, Line 2 of the second paragraph.

“Graphene is selected because of its light weight, sheet-like structure and excellent
photothermal property, while agar and silk fibroin are chosen due to their favorable
hygroscopic properties (agar and silk fibroin)^{44,45} and adhesive nature (silk fibroin),⁴⁶ which

could not only facilitate the light actuation process but also help to maintain the integrity of the
photoactuator.”.

**Q3.** Is the method proposed by the authors a generally applicable way to realize the flying
motion? If so, what is the designing principle for the flyable photoactuator, the authors should
provide more discussion.

**Answer:** We thank the reviewer for pointing this out. The method reported in the current study
is a generally applicable way to realize the flying motion (i.e. the flying based on aerodynamics)
for a photoactuator. In order to achieve the aerodynamic based flying of a photoactuator, the
general designing principles are as follows: 1) the large propulsion force, 2) high frequency
response and 3) the aerodynamically favorable structure. In terms of propulsion force and
response, we propose that the liquid to gas phase transition mechanism could be helpful. It has
been reported in the literature that the liquid to gas phase transition mechanism could
simultaneously achieve large deformation, fast response speed and large output force for soft
actuators (Nat. Comm. 2020, 11, 3988). The involvement of such mechanism could help to
increase the driving force and improve the response. In terms of the structure design, the
photoactuator needs to have an aerodynamically favorable structure, which could be achieved
by controlling the internal structures and surface patterns. Among others, the internal structures
could facilitate the liquid to gas phase transition, whereas the surface patterns may guide the
shape deformation that leads to the formation of the aerodynamically favorable structure. The
combination of all these designing principles could lead to the successful flying motion.

**Revision in the manuscript:** Page 9, Line 8 of the second paragraph.

“This mechanism also implies that, in order to achieve the rotary flight of a photoactuator, it is
desirable to satisfy the following features: 1) sufficient propulsion force output, 2) high
frequency response, and 3) the aerodynamically favorable structure.”.

**Q4.** The authors proposed that the previously developed photoactuator could not achieve the
flying motion, but what is the reason/mechanism lies behind it? The authors should elaborate
the reason/mechanism.

**Answer:** Thank you for this comment. For rotary-flying, it requires that the actuator should
have a high rotational speed. As we have summarized in Table S2, the previous photoactuators
have relatively low rotational speeds (no more than 300 rpm), making them difficult to generate
sufficient lift force for flying.

Although it is possible to improve the response frequency or the rotational speed through
structural design, it is still challenging to simultaneously satisfy large driving force, fast
response and high frequency response, which is the main hindrance that prevents them from
flying.

**Revision in the manuscript:** Page 3, Line 10 of the second paragraph.

“It is still challenging to achieve the flying locomotion mode for the photoactuators because of
their slow response speeds, small actuation force output and low response frequency.”.

Page 6, Line 9:

“The ultrafast rotation in the current study thus lays the foundation for the flight behavior. It is
in sharp contrast to the previously developed photoactuators which have relatively low
rotational speeds (no more than 300 rpm), making them difficult to generate sufficient lift force
for flying (Supplementary Table 2).”.

**Q5.** The author should give an explanations and descriptions in details about what other
researchers have done in order to improve the response/time?

**Answer:** Thank you for your comment. In order to improve the response/time of the
photoacutator, researchers have developed several strategies.

**Table R1.** Strategies that have been developed to improve the response of the photoactuators.

Photoactuator material	Shape	Response	Mechanism	Ref.
Liquid crystal gels (LCGs)	Strip	>1 s	Phase transition (in water)	1
SWNT/VO ₂ based actuators	Cantilever	3.3 ms	Phase transition	2
Iron oxide nanoparticle and poly(sodium acrylate) hydrogel actuator	Sphere	800 ms	Synergetic interactions	3
Ink/polyethylene terephthalate (PET)/acrylic actuator	Strip	360 ms	Synergetic interactions	4

**References**

- 1. Shahsavan, H., Aghakhani, A., Zeng, H., Guo, Y., Davidson, Z. S., Priimagi, A. & Sitti, M.
 Bioinspired underwater locomotion of light-driven liquid crystal gels. *Proc. Natl. Acad. Sci.*
 *U. S. A.* **117**, 5125-5133 (2020).
- 2. Wang, T., Torres, D., Fernandez, F. E., Green, A. J., Wang, C. & Sepulveda, N. Increasing
 efficiency, speed, and responsivity of vanadium dioxide based photothermally driven
 actuators using single-wall carbon nanotube thin-films. *ACS Nano* **9**, 4371-4378 (2015).
- 3. Li, M., Wang, X., Dong, B. & Sitti, M. In-air fast response and high speed jumping and
 rolling of a light-driven hydrogel actuator. *Nat. Commun.* **11**, 3988 (2020).
- 4. Li, J., Zhang, R., Mou, L., Jung de Andrade, M., Hu, X., Yu, K., Sun, J., Jia, T., Dou, Y.,
 Chen, H., Fang, S., Qian, D. & Liu, Z. Photothermal bimorph actuators with in-built cooler

for light mills, frequency switches, and soft robots. *Adv. Funct. Mater.* **29**, 1808995 (2019).

As summarized in Table R1, the utilization of the low phase temperature materials has
been demonstrated to be an effective way to improve the response/time. For example, the liquid
crystal gel exhibits a low phase transition temperature (57 °C) so that it shows almost 30 times
higher photothermal response than that of the pristine liquid crystal networks (Proc. Natl. Acad.
Sci. USA 2020, 117, 5125). In addition, the responsivity of the phase-change material (VO₂
with the low phase transition temperature of 68 °C) based photoactuator could be further
improved by 36 % when introducing materials with increased light absorbing ability and
decreased thermal conductivity (ACS Nano 2015, 9, 4, 4371).

On the other hand, the synergetic interactions are frequently exploited to further enhance
the response. For instance, the elasticity of the hydrogel and the photothermally induced phase
transition could be combined, leading to the fast response (only 800 ms) of the hydrogel in air
(Nat. Comm. 2020, 11, 3988). Furthermore, the synergistic effect between photothermal
expansion and water desorption has been utilized to increase the response of the trilayered
photoactuator to 360 ms (Adv. Funct. Mater. 2019, 29, 1808995).

**Revision in the manuscript:** Page 3, Line 9 of the second paragraph.

“Although significant progress has been made to improve the response of photoactuators
(Supplementary Table 1).”

**Added Table:** Table S1 on Page 4 in the Supplementary Information. And the corresponding
descriptions have been added below Table S1.

“In order to improve the response of the photoacutator, researchers have developed several
strategies. Among others, the utilization of the low phase temperature materials has been
demonstrated to be an effective way to improve the response. For example, the liquid crystal
gel exhibits a low phase transition temperature (57 °C) so that it shows almost 30 times higher
photothermal response than that of the pristine liquid crystal networks.¹ In addition, the

response of the phase-change material (VO_2 with the low phase transition temperature of 68°C)
 based photoactuator could be further improved by 36 % when introducing materials with
 increased light absorbing ability and decreased thermal conductivity.² On the other hand, the
 synergetic interactions are frequently exploited to further enhance the response. For instance,
 the elasticity of the hydrogel and the photothermally induced phase transition could be
 combined, leading to the fast response (only 800 ms) of the hydrogel in air.³ Furthermore, the
 synergistic effect between photothermal expansion and water desorption has been utilized to
 increase the response of the trilayered photoactuator to 360 ms.⁴”.

**Q6.** According to the mechanisms shown in the manuscript, the lift force should be a little
 different in the case of Fig. S30a and b. But the forces look likes the same, the authors should
 give a more reasonable scheme; And scale bars should be provided in Fig. S9c-j.

**Answer:** Thank you for your comments. We think the reviewer is referring to the issues in Fig.
 S30b and c (Fig. S35 in current version). We have corrected these in the revised manuscript.
 As shown in Figure R1, the lift force (F_{Lift} , red arrow) in Figure R1b is larger than that in Figure
 R1c due to the different elevation angle.

**Figure R1.** (a) The landing distance as a function of the elevation angle of the ‘helicopter’-like
 photoactuator. Force analysis of the ‘helicopter’-like photoactuator with elevation angle of (b)
 5° and (c) 25° , respectively. Error bars denote the standard deviation.

And we have added the scale bars to Fig. S9c-j, as shown in Figure R2.

**Figure R2.** (a) Schematic illustration indicating the experimental setup for the verification of
the jet propulsion. Time lapse images obtained from Supplementary Video 3 showing the
graphene/agar/silk fibroin photoactuator film (b,c) before and (d-j) after light actuation.

**Revision in the manuscript:** We have revised Supplementary Fig. S9 and S35 accordingly.

**Q7.** In the conclusion part, the authors should provide some future perspective based on the
works they have done in the manuscript.

**Answer:** Thank you for this constructive suggestion.

The flyable photoactuators may have a wide range of applications. Among others, they
could be utilized as flyable colorimetric sensors for large area environmental monitoring (based
on color analysis of digital images). Compared to the passive flying, the photoactuator in the
current study could combine the advantages from the active flying (actuated by light) and
passive wind dispersal. Since the flexible micro-electronics might also be incorporated into the

photoactuators, the distributed information may be collected remotely through wireless
modules over a long distance. In addition, the photoactuators may also have great potential in
soft robotics and other miniature devices that require the flying function.

**Revision in the manuscript:** Page 20, Line 5.

“Specifically, it is anticipated that the photoactuator reported in this study could be utilized for
the distributed environmental sensing and information collection based on either high
resolution aerial digital imaging or wireless links (through the integration of wireless modules)
over a large area.”.

**Reviewer 2**

In this manuscript, Wang et al report on the bio-inspired rotary flight of composite films, driven
by a photoactuator mechanism. The composite consists of a hygroscopic agar/silk fibroin film
that embeds graphene. Upon exposure to near-infrared light, water in the film evaporates,
leaves the film from the side and thereby generates a jet propulsion that drives the spinning of
the film (up to 7200 rpm). Due to the aerodynamic shape of the film, this results in a ‘helicopter-
like’ lift. By changing the angle at which the NIR light hits the film, the flight direction of the
film can be directed. For a film with mm dimensions, the authors report heights and horizontal
displacements in the centimetre-regime.

The work is original, the paper is well-written and the suggested mechanism for the flight
behavior is well supported by experiments. However, the following points need to be addressed
prior to publication in Nature Comm.

**Answer:** We are grateful for these comments and constructive suggestions about this work.
We have carefully addressed these issues and made proper revisions to the manuscript.
Modifications are highlighted with blue font.

**Q1.** Clarification of the mechanism: from the schemes in the main text, the driving force for
the flight is not very clear. Including a scheme like Figure S26a in the main text would help to
explain the left-vs right handed rotation. Also, the role of the microchannels in the twisting of
the sheet is not clarified in the schemes.

**Answer:** Thank you for your comments. To highlight the driving force, we have added the top
view scheme that indicates the driving force (F_{jet}) to Figure R3e, as shown in the following:

**Figure R3.** (a-f) Schematic illustrating the flight mechanism of the 'helicopter'-like
 photoactuator.

As can be seen from the top view scheme shown in Figure R3e, both the driving force and
 the rotation direction are illustrated.

In order to clearly explain the left-vs right handed rotation, we have also added the top
 view schemes that show the driving force to Figure R4a and Figure R4b, as shown in the
 following:

**Figure R4.** Overlaid CCD images showing the 'helicopter'-like photoactuator flying to (a) the
 left rear and (b) the right rear, which is realized by controlling the elevation angle. The
 corresponding top view schemes indicating the driving force and rotational direction are shown
 in the insets in (a) and (b), respectively.

In addition, we have altered the color of the microchannels in the schemes to make them
 more clear so that their roles in twisting the film are better illustrated, as shown in Figure R3
 and Figure R4.

**Revised Figure:**

Figure R3 has been added to Figure 2 as Figure 2a-f, Figure R4 has been added to Figure 3 as
Figure 3e and 3j on Page 33,34.

**Q2.** Reusing the films for multiple flights: The authors mention that the films can fly multiple
258 times. However, the underlying mechanism relies on the evaporation of water upon heating of
259 the film. Hence, the spinning stops when all water is evaporated-which is within a couple of
260 seconds, implying that the films can only fly short distances. How long does it take for a film
to re-absorb water, such that it will fly again upon new exposure to NIR?

**Answer:** Thank you for your comments. We have examined the water re-absorption time after
light actuation for 7 cycles, as shown below:

**Figure R5.** Water re-absorption time of the ‘helicopter’-like photoactuator after repetitive
actuation. Error bars denote the standard deviation.

As can be seen from Figure R5, the average water re-absorption time is approximately 5.5
270 min.

**Revision in the manuscript:** Page 11, Line 5 of the first paragraph.

“We have also examined the water re-absorption time after light actuation for 7 cycles. As

shown in Supplementary Figs. 18 and 19, it takes approximately 5.5 min to re-absorb water
after repetitive light actuation and the local heating does not affect the capability of the film to
re-absorb water.”.

**Added Figure:**

Figure R5 has been added as Supplementary Fig. 18 on Page 15 in the Supplementary
Information.

**Q3.** Also, the authors mention that they inspected the film after the actuation process. The
morphologies of the protrusion surface and the film are shown to be unaffected. However, does
the local heating of the film affect the capability of the film to re-absorb water?

**Answer:** Thank you for pointing this out. In order to study whether the local heating of the film
affects the capability of the film to re-absorb water, we have tested the weight of the
photoactuator film before aviation, after aviation, and after water re-absorption.

**Figure R6.** Weight variation of the photoactuator film before aviation (red), after aviation (pink)
and after water re-absorption (blue).

As shown in Figure R6, the photoactuator film loses about 9.5 % water after light-driven

aviation. After water re-absorption by exposing the film to the environment with 90 % relative
humidity, the weight of the film could be recovered. In the above experiment, we have
examined 7 re-absorption cycles and the water absorption capability of the photoactuator film
remains nearly unchanged.

**Revision in the manuscript:** Page 11, Line 5 of the first paragraph.

“We have also examined the water re-absorption time after light actuation for 7 cycles. As
shown in Supplementary Figs. 18 and 19, it takes approximately 5.5 min to re-absorb water
after repetitive light actuation and the local heating does not affect the capability of the film to
re-absorb water.”.

**Added Figure:**

Figure R6 has been added as Supplementary Fig. 19 on Page 15 in the Supplementary
Information.

**Q4.** Could it be that the local heating generates a “preferred spot” to create the jet propulsion
opening when the film is re-exposed to NIR? This might affect the capabilities to steer the film
during subsequent flights.

**Answer:** We thank the reviewer for pointing this out. We agree with the reviewer that the local
heating could generate a ‘preferred spot’ when the film is re-exposed to the NIR light. As can
be seen from the following figure (Figure R7), after light actuation, the white powders are
blown away basically at the same position for the first two cycles, which indicates the jetting
position may be unchanged.

**Figure R7.** The images showing the photoactuator film after (a) first and (b) second light
 actuation. The generated jet can be directly visualized by the blown white powders.

In order to confirm whether there is a ‘preferred spot’ at the micron scale, we have
 examined the film edge where the jet propulsion occurs under optical microscope, as shown
 below.

**Figure R8.** Microscopic images showing the cross-section of the film edge where the jet
 propulsion occurs after (a) first and (b) second light actuation. Note that there is a marker which
 is formed by cutting the film edge prior to the light actuation.

Prior to the light actuation, we have cut the film edge so that the cut could be utilized as a
marker (Figure R8). After light actuation, the distance from the marker to the position where
the jet propulsion occurs is measured. The distance is approximately 1 mm after the first light
actuation and remains unchanged after the second cycle of light actuation. In addition, we have
measured the distance from the marker to the jetting position after 7 actuation cycles, as shown
in the following figure (Figure R9). The distance variation is negligible. This further indicates
that, after the initial jetting, a ‘preferred spot’ forms at the film edge, and the water vapor
consistently jets from this ‘preferred spot’ during the subsequent cycles.

**Figure R9.** Distance from the marker to the ‘preferred spot’ after light actuation for 7 cycles.

**Revision in the manuscript:** Page 11, Line 8 of the first paragraph.
“In addition, it is worth pointing out that the local heating could generate a ‘preferred spot’ at
which the jet propulsion occurs upon repetitive light actuation (Supplementary Figs. 20-22).”.

**Added Figures:**
Figure R7-9 has been added as Supplementary Figs. 20-22 on Page 16,17 in the Supplementary
Information.

**Q5.** Minor point: The paper mentions nanocomposite elements, but is it really nanocomposite?

The films appear to me as if it consists of multi-layered graphene embedded in the composite,
without control at the nano-level (i.e. single sheet graphene).

**Answer:** We do appreciate this constructive suggestion. We have carefully considered the
definition of the ‘nanocomposite’ and agreed with the reviewer. As suggested, we have
changed all the ‘nanocomposite film’ to ‘photoactuator’ in the manuscript for clarity.

**Reviewer 3**

I generally liked the idea of the paper and found the experimental realization very funny. I
assume the authors had a very enjoyable project. The science seems solid and well-done.

**Q1.** The paper itself, however, is not really at a level required for Nature Communication.
While parts of the manuscript are well-written, in correct English and clearly structured, others
are not. The beginning of the introduction repeats phrased and wordings, without paying
intention to grammar, the use of articles... It is clear that different people (with different
linguistic skill levels) wrote the article. In this form, it should not be published without a
thorough language editing.

**Answer:** Thank you very much for your constructive comments. After carefully examining the
manuscript, we agree with the reviewer and are fully aware of the problems the reviewer has
mentioned, including, but not limited to, the structures, wordings, grammars, etc. In order to
make the manuscript more readable, we have thoroughly revised the manuscript according to
the reviewer's comments. Revisions are highlighted with blue font in the manuscript.

**Q2.** Also, I find the application part absolutely not convincing, why is it impressive that from
1.5 m starting height in presence of wind the helicopters are distributed about 1.1 m?

**Answer:** We are thankful for this comment. After careful consideration, we think the
description in the section heading, i.e. "Potential application", is inaccurate and have changed
it to "Potential application demonstration".

In nature, many plants use wind to spread their seeds over a large area. Inspired by nature,
researchers have developed wind-dispersal microfliers with on-board sensors (colorimetric
sensors (read out based on color analysis of the digital images) or electronic ones (read out
through wireless links), which could distribute themselves over a large area to collect collective

information for environmental monitoring or sensing purposes (Nature 2022, 603, 427; Nature
2021, 597, 503).

In the ‘Potential application demonstration’ section of the revised manuscript, as the shape
of the photoactuator is similar to that of the vine maple seed, we have first studied its wind-
dispersal behavior and compared it to that of the plant counterpart. Second, we have examined
the parameters that may influence the wind-dispersal behavior of the photoactuator. Third, as
a proof-of-the-concept example, we have explored the potential of the photoactuators as wind-
dispersed colorimetric sensors for collective environmental information collection.

We have first fabricated a photoactuator with a size similar to that of the vine maple seed
(80 mm^2). Depending on the varieties of vine maple trees, their heights range from less than 1
398 m to over 4 m. The seeds of the vine maple tree in our campus distribute randomly on the tree
with heights ranging from $\sim 0.5 \text{ m}$ to $\sim 2.5 \text{ m}$ (averaging 1.5 m). We thus choose to release the
photoactuator at a similar height, i.e. 1.5 m.

We have compared the wind-dispersal distance of the photoactuator film with and without
the ‘airscrew’-like structure.

**Figure R10.** The wind-dispersal distance of the composite film with (red column) and without
(orange column) the ‘airscrew’-like structure. Error bars denote the standard deviation.

As shown in Figure R10, the wind-dispersal distance of the composite film with the
‘airscrew’-like structure (red column) is significantly farther than that without the ‘airscrew’-

like structure (orange column), revealing the importance of the ‘airscrew’-like structure. This
is because the composite film without the ‘airscrew’-like structure tumbles during the falling
process (A new video is recorded and shown in the supplementary information as Video S10),
while the composite film with the ‘airscrew’-like structure could maintain the stability during
the flying motion which could facilitate its potential application. Since plants are known to use
wind to efficiently disperse their seeds, we have compared the wind-dispersal performance of
the photoactuator with that of the vine maple seed. Under the influence of 3 m/s wind, the
photoactuator film could be wind-dispersed to a distance of 1.1 m, which is similar to that of
the winged vine maple seed (i.e. 1.05 m) under the similar circumstance, further indicating the
effectiveness of the ‘airscrew’-like structure.

Second, we have investigated the influence of the release height and the size of the
photoactuator on the wind-dispersal distance.

**Figure R11.** Wind-dispersal distance of the photoactuator released from different height. Error
bars denote the standard deviation.

As shown in the above figure (Figure R11), the photoactuator can be wind-dispersed to a
farther distance when the release height is increased, and the maximum dispersal distance can
reach approximately 10 m (the release height and the wind speed are 5 m and 3 m/s,
respectively).

We have also examined the influence of the size of the photoactuator (the release height

and the wind speed are 1.5 m and 3 m/s, respectively) on the wind-dispersal distance.

**Figure R12.** The effect of the size of the photoactuator on its wind-dispersal distance (the
release height and the wind speed are 1.5 m and 3 m/s, respectively). Error bars denote the
standard deviation.

As can be seen from Figure R12, there is a maximum wind-dispersal distance (2.2 m) when
the size of the photoactuator is about 20 mm². This is possibly because the size of the
photoactuator should be large enough to generate sufficient lift force. However, as the size
increases, the weight of the photoactuator also increases, which may result in the decrease of
the flying ability.

Third, as a proof-of-the-concept example, we have explored the potential of the
photoactuators in collective colorimetric sensing for environmental monitoring. As shown in
Figure R13, the photoactuator is loaded with four thermosensitive gels and four humidity
sensitive gels, which could be utilized to collect the temperature and humidity information,
respectively (Figure R13a-c). A transparent polydimethylsiloxane (PDMS) substrate is also
attached onto the photoactuator to collect the particle matter for pollutant measurement. For a
single photoactuator, after actuation, the information about temperature and relative humidity
could be visually read out through the color changes of thermosensitive and humidity sensitive
gels (Figure R13d,e) in the captured digital image. The size of the particulate pollutant could
also be obtained by analyzing the particle matter on the PDMS substrate (Figure R13f-h) based

on the dynamic light scattering (DLS) method. Therefore, by combining the flying motion and
 the wind-dispersal capability of the photoactuator, it may be possible to achieve the collective
 sensing in complex environments.

 **Figure R13.** (a) CCD image showing the photoactuator integrated with a PDMS substrate, four
 thermosensitive gels and four humidity sensitive gels. (b) Colorimetric change of the humidity
 sensitive gel at different humidity. (c) Colorimetric change of the thermosensitive gel at
 different temperature. CCD images showing the photoactuator with surface-attached sensors
 and PDMS substrate (d) before and (e) after light actuation. Optical microscopic images of the
 surface-attached PDMS substrate (f) before release and (g) after landing (the release height and

the wind speed are 1.5 m and 3 m/s, respectively). (h) Size distribution of the particulate
pollutant.

To this end, we have set up four areas with different environmental characteristics,
representing hot, cold, humid and dusty environments (as shown in Figure R14). To monitor
the environmental characteristics of different areas, we have controlled the flight of eight
photoactuators (Number 1 to 8) toward different regions (the release height and the wind speed
are 1.5 m and 3 m/s, respectively). By direct color analysis of the attached colorimetric sensors
on the photoactuators (based on the digital image), we could simultaneously collect the
environmental information from different regions (Figure R14). Additionally, it is also possible
to evaluate the particulate pollutant level based on the DLS measurement. Based on this method,
the distribution of the temperature (Figure R15a), humidity (Figure R15b) and size of the
particulate pollutant (Figure R15c) over the whole area (shown in Figure R14) could be
obtained.

**Figure R14.** Digital image of the distributed photoactuators for collective environmental

monitoring. The hot, cold, humid and dusty areas are all indicated. The corresponding enlarged
images at different locations (from 1 to 8) are also shown.

**Figure R15.** (a) Temperature distribution in the area shown in Supplementary Fig. S42
obtained based on the color analysis of the colorimetric temperature sensors on the landed
photoactuators. (b) Humidity distribution in this area obtained based on the color analysis of
the colorimetric humidity sensors on the landed photoactuators. (c) Size distribution of
particulate pollution in this area which is obtained by analyzing the PDMS substrates on the
surface of the landed photoactuators based on the dynamic light scattering (DLS) method.

Compared to the traditional sensors for environmental monitoring, the photoactuator based
ones may offer several advantages:

1. The distributed collections of multiple sensors allow for simultaneous and collective
sensing to obtain information from different areas.

2. The active light actuation enables the flying locomotion with the guided direction and the
passive wind-dispersal could increase the flying distance, both of which make the
photoactuators suitable for collective sensing.

3. The ‘airscrew’-like structure makes the flying stable without tumbling, which could
facilitate the distributed sensing.

**Revision in the manuscript:** Page 16, Line 11.

“We have further compared the wind-dispersal distance of the ‘helicopter’-like photoactuator
with and without the ‘airscrew’-like structure. As shown in Supplementary Fig. 38, the wind-
dispersal distance of the composite film with the ‘airscrew’-like structure (red column) is
significantly farther than that without the ‘airscrew’-like structure (orange column), revealing
the importance of the ‘airscrew’-like structure. This is because the composite film without the
‘airscrew’-like structure tumbles during the falling process (Supplementary Video 10), while
the composite film with the ‘airscrew’-like structure could maintain the stability during the
flying motion. Furthermore, the release height and size of the photoactuator could also
influence the wind-dispersal distance. As shown in Supplementary Fig. 39, the photoactuator
can be wind-dispersed to a farther distance when the release height is increased, and the wind-
dispersal distance can reach approximately 10 m when the release height is 5 m (the wind speed
is 3 m/s). In addition, as can be seen from Supplementary Fig. 40, there is a maximum wind-
dispersal distance (2.2 m) when the size of the photoactuator is about 20 mm². This is possibly
because the photoactuator size should be large enough to generate the lift force. However, as
the size increases, the weight of the photoactuator also increases, which may result in the
decrease of the flying ability.”.

Page 17, Line 1 of the second paragraph.

“Inspired by nature, researchers have developed wind-dispersal microfliers with on-board
sensors (colorimetric sensors (read out based on color analysis of the digital images) or
electronic ones (read out through wireless links)), which could distribute themselves over a
large area to collect collective information for environmental monitoring or sensing purposes.”.

Page 17, Line 8 of the second paragraph.

“As a proof-of-the-concept, we have explored the potential of the photoactuators in collective
colorimetric sensing for environmental monitoring. As shown in Supplementary Fig. 41a, the
photoactuator is loaded with four thermosensitive gels and four humidity sensitive gels, which
could be utilized to collect the temperature and humidity information, respectively. A
transparent polydimethylsiloxane (PDMS) substrate is also attached onto the photoactuator in
order to collect the particle matter for pollutant measurement. After light actuation and wind-
dispersal, the information about temperature and relative humidity could be visually read out
through the color changes of the thermosensitive and humidity sensitive gels (Supplementary
Fig. 41b-e). The size of the particulate pollutant could also be obtained by analyzing the particle
matter on the PDMS substrate (Supplementary Fig. 41f-h) based on the dynamic light
scattering (DLS) method. Therefore, by combining the flying motion and the wind-dispersal
capability of a number of photoactuators, it is possible to achieve the collective sensing in the
complex environment.

To this end, we have set up four areas with different environmental characteristics,
representing hot, cold, humid and dusty environments (Supplementary Fig. 42). To monitor the
environmental characteristics of different areas, we have controlled the flight of eight
photoactuators (Number 1 to 8) to fly toward different regions (the release height and the wind
speed are 1.5 m and 3 m/s, respectively). By directly visualizing the attached colorimetric
sensors on the photoactuators based on the captured digital image, we could simultaneously
collect the environmental information from different regions. Additionally, it is also possible
to evaluate the particulate pollutant level based on the DLS measurement. Based on this method,
the distribution of temperature, humidity and size of the particulate pollutant (Supplementary
Fig. 43) over the whole area could be obtained. Compared to the traditional sensors for
environmental monitoring, the photoactuator based ones may offer several advantages: 1) The
distributed collections of multiple sensors allow for simultaneous and collective sensing to
obtain information from different areas; 2) The active light actuation enables the flying
locomotion with the guided direction and the passive wind-dispersal could increase the flying
distance, both of which make the photoactuators suitable for collective sensing; 3) The

'airscrew'-like structure makes the flying stable without tumbling, which could also facilitate
the distributed sensing.”.

**Added Figures:**

Figure R10-15 has been added as Supplementary Fig. 38-43 on Page 24-28 in the
Supplementary Information.

We hope that our replies and the revised manuscript are satisfactory to you. We will be
happy to provide any further information if needed.

REVIEWERS' COMMENTS

Reviewer #1 (Remarks to the Author):

The current version after revision could be accepted as is.

Reviewer #2 (Remarks to the Author):

The authors have properly revised their manuscript, regarding both my own comments and suggestions as well as those of the other reviewers, and I recommend the manuscript for Nature Comm.

I have only two minor comments:

Figure S42: The positioning of the humidity and temp. sensitive gels at the photoactuator strip (i.e. top vs. bottom) seems to be different for the different samples 1-8, which is confusing.

Materials and Methods: what type of thermo- and humidity sensitive gels were used?

Reviewer #3 (Remarks to the Author):

I acknowledge the thorough revision of the paper and find it now suitable for publication.

We have carefully considered the reviewers' comments and revised the manuscript accordingly, detailed as follows:

Reviewer 2

Comments: The authors have properly revised their manuscript, regarding both my own comments and suggestions as well as those of the other reviewers, and I recommend the manuscript for Nature Comm. I have only two minor comments:

Q1. Figure S42: The positioning of the humidity and temp. sensitive gels at the photoactuator strip (i.e. top vs. bottom) seems to be different for the different samples 1-8, which is confusing.

Answer: We thank the reviewer for pointing this out. In order to demonstrate both clockwise and counterclockwise rotational films could be successfully deployed, we have released six films (i.e. film No. 1,2,3,5,7,8) with clockwise rotation and two films (i.e. film No. 4,6) with counterclockwise rotation (totally eight films). On the surface of these films, there are two rows of sensitive gels. The row consisting of humidity sensitive gels is placed closer to the jet propulsion position. The row consisting of thermosensitive gels is placed farther to the jet propulsion position. In addition, because of the large occupied surface area of the group containing the entire six sensitive gels (three humidity sensitive gels and three thermosensitive gels), they are placed to the right of the jet propulsion position in case of the clockwise rotational flying and to the left of the jet propulsion position in the case of the counterclockwise rotational flying in order to realize the success flight.

To make Supplementary Fig. 42 more clear, we have added more descriptions in the

29 figure caption, as shown in the following:

**Supplementary Fig. 42.** Digital image of the distributed photoactuators for collective
environmental monitoring. The hot, cold, humid and dusty areas are all indicated. The
corresponding enlarged images at different locations (from 1 to 8) are also shown. In order to
demonstrate both clockwise and counterclockwise rotational films could be successfully
deployed, we have released six films (i.e. film No. 1,2,3,5,7,8) with clockwise rotation and
two films (i.e. film No. 4,6) with counterclockwise rotation (totally eight films). On the
surface of these films, there are two rows of sensitive gels. The row consisting of humidity
sensitive gels is placed closer to the jet propulsion position. The row consisting of
thermosensitive gels is placed farther to the jet propulsion position. In addition, because of
the large occupied surface area of the group containing the entire six sensitive gels (three
humidity sensitive gels and three thermosensitive gels), they are placed to the right of the jet
propulsion position in case of the clockwise rotational flying and to the left of the jet
propulsion position in the case of the counterclockwise rotational flying in order to realize the

success flight.

**Q2.** Materials and Methods: what type of thermo- and humidity sensitive gels were used?

**Answer:** We thank the reviewer for pointing this out. The thermosensitive gel is made up of
the leuco dye-developer (dodecyl gallate, etc.)-solvent (octadecanol, etc.) system which is
embedded in the epoxy polymer matrix. The humidity sensitive gel consists of the silica gel
and the color-changing indicator (i.e. cobalt chloride).

**Revision in the manuscript:** Page 21, Line 6 of the second paragraph.

“The thermosensitive gel consisting of the leuco dye-developer (dodecyl gallate, etc.)-solvent
(octadecanol, etc.) system embedded in the epoxy polymer matrix was obtained from
Shenzhen MiJiang Tech. Co., Ltd. The humidity sensitive gel containing the silica gel and the
color-changing indicator (i.e. cobalt chloride) was purchased from Dongguan Zhongqi
Printing Equipment Co., Ltd.”.

Modifications in the manuscript are highlighted with blue font.

We hope that our replies and the revised manuscript are satisfactory to you. We will be
happy to provide any further information if needed.
